# Cryo-EM structure of the bacterial Ton motor subcomplex ExbB–ExbD provides information on structure and stoichiometry

Herve Celia [1], Istvan Botos[1], Xiaodan Ni[2], Tara Fox[3,4], Natalia De Val[3,4], Roland Lloubes[5], Jiansen Jiang [2]* & Susan K. Buchanan [1]*

The TonB–ExbB–ExbD molecular motor harnesses the proton motive force across the bacterial inner membrane to couple energy to transporters at the outer membrane, facilitating uptake of essential nutrients such as iron and cobalamine. TonB physically interacts with the nutrient-loaded transporter to exert a force that opens an import pathway across the outer membrane. Until recently, no high-resolution structural information was available for this unique molecular motor. We published the first crystal structure of ExbB–ExbD in 2016 and showed that five copies of ExbB are arranged as a pentamer around a single copy of ExbD. However, our spectroscopic experiments clearly indicated that two copies of ExbD are present in the complex. To resolve this ambiguity, we used single-particle cryo-electron microscopy to show that the ExbB pentamer encloses a dimer of ExbD in its transmembrane pore, and not a monomer as previously reported. The revised stoichiometry has implications for motor function.

[1] Laboratory of Molecular Biology, National Institute of Diabetes and Digestive and Kidney Diseases, National Institutes of Health, Bethesda, MD 20892, USA. [2] Laboratory of Membrane Proteins and Structural Biology, Biochemistry and Biophysics Center, National Heart, Lung, and Blood Institute, National Institutes of Health, Bethesda, MD 20892, USA. [3] Center for Molecular Microscopy, Center for Cancer Research, National Cancer institute, National Institutes of Health, Bethesda, MD 20892, USA. [4] Cancer Research Technology Program, Frederick National Laboratory for Cancer Research, Leidos Biomedical Research Inc, Frederick, MD 21701, USA. [5] Laboratoire d'Ingénierie des Systèmes Macromoléculaires, UMR7255 CNRS/Aix-Marseille Université, Institut de Microbiologie de la Méditerranée, 13402 Marseille Cedex 20, France. *email: jiansen.jiang@nih.gov; susan.buchanan2@nih.gov

Ton is a membrane protein complex found in the inner membrane of Gram-negative bacteria. Ton can be described as a molecular motor in which the ExbB and ExbD subunits assemble to form a proton channel and use the proton gradient across the cytoplasmic membrane. The energy derived from proton translocation is propagated through the elongated TonB subunit that spans the periplasmic space and physically interacts with numerous TonB dependent transporters (TBDT) at the outer membrane, allowing the active transport of rare nutrients into the periplasm[1]. TonB (239 residues) and ExbD (141 residues) have a similar topology: a short N-terminal domain in the cytoplasm, followed by a single transmembrane (TM) helix, and a periplasmic domain composed of a flexible linker and a folded C-terminal domain[2–4]. ExbB (244 residues) has three TM helices, with a short periplasmic N-terminal domain, a small periplasmic loop between TM 2 and 3, and a large cytoplasmic domain[5]. We previously reported the crystal structure of ExbB in complex with a truncated form of ExbD that lacked the periplasmic domain (ExbD$_{deltaperi}$). The crystal structure shows a pentamer of ExbB containing a TM hydrophobic pore that sits 10 Å above the membrane plane, in which one TM helix of the ExbD subunit was found; however, we observed two copies of ExbD by cysteine crosslinking and DEER experiments[5]. Based on these results, we proposed a model for the Ton subcomplex where the ExbB pentamer encloses one copy of ExbD in the pore, with a second copy of ExbD sitting outside the pentamer.

To gain further information on structure and stoichiometry, we performed a single particle cryo-electron microscopy (cryo-EM) analysis on the full-length ExbB–ExbD complex reconstituted in lipid nanodiscs. The new structure shows that five copies of ExbB are arranged as a pentamer around two copies of ExbD, which sit in the pore. Compared to the crystal structure, the two copies of ExbD sit about 3 Å higher in the pore indicating that vertical, piston-like movement is a possible mode of action. The five copies of ExbB are shifted outward to accommodate both copies of ExbD, with no proton channel available in this conformation. Disulfide crosslinking experiments at various locations on ExbD provide supporting evidence for the observed stoichiometry and arrangement.

## Results

**Cryo-EM structure of ExbB–ExbD in lipid nanodiscs**. The final reconstruction used 85,936 particles and yielded a 3.3 Å overall resolution structure calculated without imposing symmetry (Table 1; Supplementary Figs 1–3). The structure reveals numerous alpha helices, with clear side chain densities and secondary structure elements, that allowed an atomic model to be built (Fig. 1). As found in the crystal structure, ExbB assembles as a pentamer, but instead of one ExbD TM domain, two TM helices of ExbD are now clearly resolved in the TM pore defined by the ExbB pentamer (Fig. 1c, d; Supplementary Fig. 2). The two TM helices are parallel but shifted relative to one another by half a helical turn. The two essential aspartate residues at position 25 point in opposite directions and are in close proximity to the ExbB conserved threonine residues 181 and 148 that form a ring of polar residues in the pore (Fig. 2a, b). The two ExbD TM helices pack tightly into the pore, with nearly 80% of surface buried for each helix, and there is no apparent channel in this configuration that would allow a proton to go through (Fig. 2c). While ExbB assembles as a pentamer, the top of the pore is larger relative to the crystal structure (Supplementary Movie 1 and 2). A comparison of the EM and crystal structures shows that the most dramatic differences for ExbB are seen in shifts of some of the TM helices to accommodate the dimer of ExbD (Supplementary Fig. 4). No density is visible for the periplasmic domain of ExbD,

### Table 1 Data collection, processing, and refinement statistics

| | |
|---|---|
| *Deposition ID* | EMD-20583; PDB 6TYI |
| *Data collection and processing* | |
| Magnification | 130,000 (nominal) |
| Voltage (kV) | 300 |
| Electron exposure (e−/Å$^2$) | 71 |
| Defocus range (μm) | −0.7 to −2.5 |
| Pixel size (Å) | 1.06 |
| Symmetry imposed | C1 |
| Initial particle images (no.) | 3,206,108 |
| Final particle images (no.) | 85,936 |
| Map resolution (Å) | 3.3 |
| FSC threshold | 0.143 |
| Map resolution range (Å) | 3.0–19.0 |
| *Refinement* | |
| Initial model used (PDB code) | 5SV0 |
| Map sharpening *B* factor (Å$^2$) | −93.6 |
| Model composition | |
| Non-hydrogen atoms | 9478 |
| Protein residues | 1186 |
| Ligands | 4 |
| *B* factors (Å$^2$) | |
| Protein | 45.51 |
| Ligand | 44.60 |
| R.m.s. deviations | |
| Bond length (Å) (# > 4σ) | 0.012 (10) |
| Bond angles (°) (# > 4σ) | 0.901 (1) |
| Validation | |
| Refined model CC | 0.82 |
| MolProbity score | 1.83 |
| Clashscore | 8.55 |
| Poor rotamers (%) | 0.55 |
| Ramachandran plot | |
| Favored (%) | 94.71 |
| Allowed (%) | 5.29 |
| Disallowed (%) | 0 |

likely because the 19-residue linker that connects the TM to the folded C-terminal domain is highly flexible[4]. Four phospholipids were built into grooves defined by the hydrophobic portions of the α6 and α7 TM helices of two adjacent ExbB subunits. The polar head groups of the phospholipids point toward the lateral fenestrations of the complex that are lined up with basic residues (Supplementary Fig. 5).

**Cysteine crosslinking shows two copies of ExbD in the pore**. To verify whether this new structure is representative of the whole population of ExbB–ExbD complexes, we performed cysteine crosslinking experiments and introduced single cysteines at three different positions on ExbD, either in the cytoplasm (D10C), the TM region (L40C) or the linker region between the TM and C-terminal domain (P50C). Co-expression and purification with ExbB depleted of cysteine show efficient disulfide crosslink formation for all three constructs (Fig. 2d). SEC experiments confirm that the disulfide bonds occur within the same ExbB–ExbD complex (Fig. 2e, f). While efficient crosslinking with P50C is expected for both configurations, i.e. one or two ExbD TMs in the pore, crosslinking with D10C or L40C can only occur if both the TM and N-terminal domains of ExbD are located in the pentameric pore and cytoplasmic cavity of ExbB (Fig. 2e). As assessed by SDS-PAGE, the crosslinking of L40C is not as efficient as that for D10C and P50C. Examination of the cryo-EM structure shows that the two Leu40 side chains are pointing in opposite directions in the pore. The fact that L40C is able to crosslink at all suggests that the ExbD TM helices are mobile and can rotate.

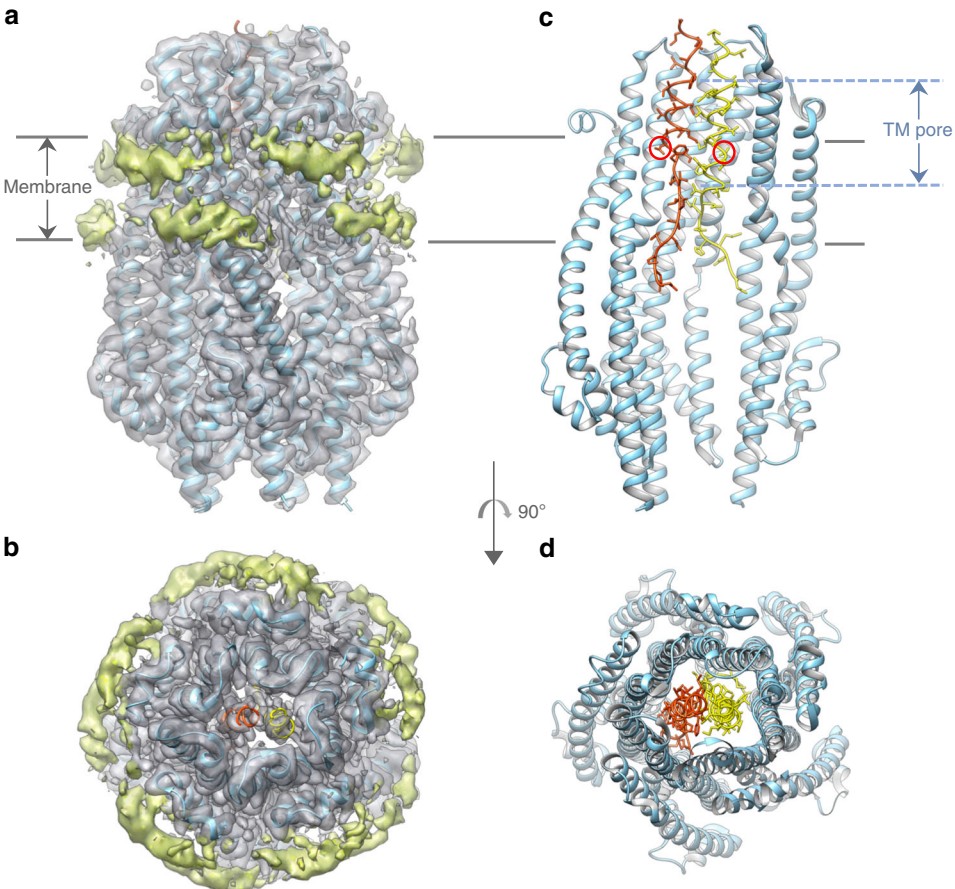

**Fig. 1** Structure of the ExbB-ExbD subcomplex in lipid nanodiscs. **a** View perpendicular from the membrane of the ExbB–ExbD cryo-EM structure represented as isosurfaces. The protein contribution is colored in gray, while the contribution of the nanodisc is colored in green. The boundary of the nanodisc allows one to estimate the position of the lipid membrane. The reconstructed atomic model is shown in ribbon representation. **b** Same as A but viewed from the periplasm. **c** Ribbon representation of the reconstructed atomic model, colored in cyan for the ExbB pentamer, red and yellow for the two TM domains of the two ExbD subunits. For clarity only three ExbB subunits are represented. The TM pore formed by the ExbB pentamer is above the membrane plane. The essential Asp25 of ExbD are shown with red circles. **d** Same as C, viewed from the periplasm

## Discussion

The cryo-EM structure shows that the pentamer of ExbB encloses a dimer of ExbD TM helices in its central pore. The exact stoichiometry of the Ton complex has always been a matter of debate, with different stoichiometries reported ranging from four ExbB and one or two ExbD, five ExbB and two ExbD, or six ExbB and three ExbD[5–7]. However, recent mass spectrometry experiments performed on native membranes of *E. coli* cells detected only the pentameric form of ExbB, suggesting that the tetramer and hexamer forms are non-physiological or of very low abundance[8].

Because of the reconstitution in lipid nanodiscs, the ExbB–ExbD subcomplex is observed in a natural lipid environment. The cryo-EM map reveals densities that correspond to phospholipids closely associated with the ExbB pentamer. Four phospholipids were built, three PE and one PG, and are part of the cytoplasmic leaflet of the membrane. The crystal structure did not reveal any associated lipids, likely because they were displaced by the detergent.

The ExbD periplasmic domain dimerizes in the ExbB–ExbD complex, and the structure of the soluble periplasmic domains of the closely related MotB and TolR homologues are dimeric[5,9,10]. It was therefore surprising to find only one TM domain of ExbD in the ExbB pentameric crystal structure, as disulfide cysteine scanning experiments with TolR or MotB suggested that these proteins also dimerize in their membrane embedded region[11,12]. It is not fully understood why only one TM domain of the

ExbD$_{deltaperi}$ construct was found in the ExbB pentameric crystallographic structure, but it is likely related to the fact that this truncated construct lacks the C-terminal periplasmic domain of ExbD, which has been shown to dimerize in vivo[13].

The observed densities in the cryo-EM structure are better defined for the ExbB subunits than for ExbD subunits, suggesting that the ExbD subunits are somewhat mobile (Supplementary Fig. 3). The cryo-EM experiment was performed at neutral pH (7.5), while the crystal structure that showed the single ExbD TM domain was obtained with crystals grown at acidic pH (4.5). It is likely that the cryo-EM structure represents the resting state of the Ton complex, as there is no visible channel that would connect the Asp25 residues either to the periplasm or cytoplasm. In the crystal structure obtained at acidic pH, the ExbD TM helix is shifted by about 3 Å compared to the cryo-EM structure, bringing Asp25 closer to the cytoplasmic cavity (Supplementary Fig. 4). As Asp25 is likely to be protonated in the crystal structure, it might represent an intermediate in which the Asp is primed to deliver a proton to the cytoplasm. If correct, this would be indicative of a translational movement along the pore axis of one or both of the ExbD TM helices during the catalytic cycle.

It is hypothesized that the proton translocation through the Ton complex results in movement propagation. The two ExbD TM helices might move up and down the pore axis, and/or rotate during the catalytic process. Inspection of the cryo-EM structure shows that the ExbD TM domains fit tightly in the pore and do

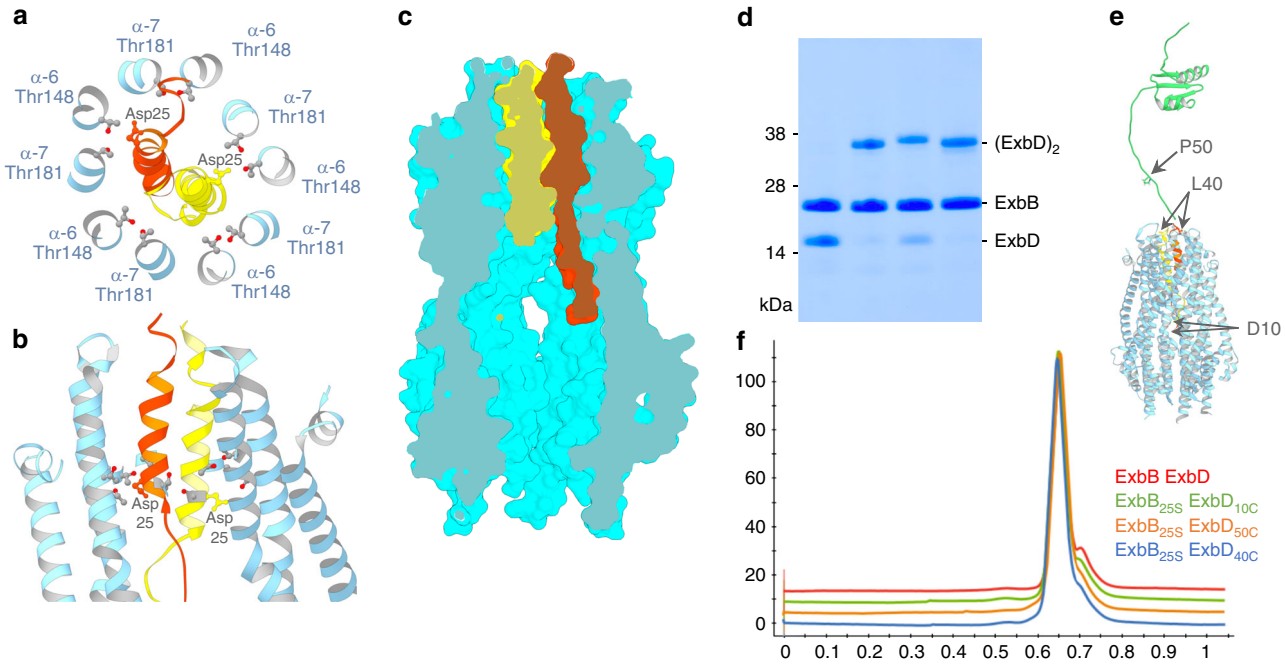

**Fig. 2** Arrangement of ExbD α-helices in the ExbB-ExbD subcomplex. **a** Ribbon representation of the TM domains of ExbD, viewed from the cytoplasm, colored in orange and yellow. The atoms of the essential Asp25 residues are represented in ball and sticks. The ring of conserved threonines 148 on ExbB TM α6 and 181 on TM α7 are shown as ball and sticks. **b** Same as A but viewed perpendicular to the membrane plane. For clarity only two ExbB subunits are shown. **c** Section through a surface representation of the ExbB pentamer (cyan) and the two ExbD TM domains (orange and yellow) **d** SDS-Page of DDM solubilized and purified ExbB–ExbD complexes. The first lane is the control with no cysteine on ExbD, the second, third and fourth lane are ExbB$_{C25S}$–ExbD$_{D10C}$, ExbB$_{C25S}$–ExbD$_{L40C}$ and ExbB$_{C25S}$–ExbD$_{P50C}$ respectively. **e** Ribbon representation of the solution structure of the ExbB soluble periplasmic domain (in green, pdb code 2PDU)[4] and ExbB–ExbD. The positions of the D10, L40, and P50 residues of ExbD are indicated with arrows. **f** SEC elution profiles of the DDM solubilized and purified ExbB–ExbD complexes. The X axis is expressed as column volume (Superose6 increase 10/30), the Y axis is in milli-absorbance units at 280 nm

not have enough space to move in either direction, unless the ExbB subunits move as well. Inspection of the cryo-EM and crystal structures shows that the ExbB pentamer can adopt different conformations (Supplementary Fig. 4, Supplementary Movies 1 and 2). When overexpressed, ExbB is also able to assemble as a hexamer and can form a stable oligomer in the absence of ExbD[7,14]. This structural plasticity might be important to accommodate the different structural states of the ExbD TM domains.

Taken together, we have demonstrated that the Ton subcomplex consists of a pentamer of ExbB enclosing a dimer of ExbD TM domains. The essential Asp25 residues on ExbD are in close proximity to the conserved Thr 148 and 181 on ExbB. Because there is no apparent channel in this structure through which a proton could be translocated, conformational changes must occur to open such a channel, possibly involving interaction with the TonB subunit. We showed previously that addition of TonB to the ExbB–ExbD subcomplex does not change the stoichiometries for ExbB and ExbD[5].

## Methods

**Cloning and purification.** Native *exbB* was cloned into pET26b, *exbD* harboring a 3′ end TEV protease site followed by 10 His codons was inserted into pCDF-1b[5]. The D10C, L40C, and P50C cysteine substitutions of ExbD and C25S mutation of ExbB were obtained by quick change PCR using specific primers (primer sequences available upon request)[15]. The sequences of all plasmid constructs were verified by sequencing analysis.

The plasmid encoding the MPS1E3D1 apo-lipoprotein was obtained from Addgene and expressed and purified as described[16]. The N-terminal histag was cleaved using TEV protease. The resulting MSP1E3D1 was concentrated to 230 μM, flash frozen in liquid nitrogen and stored at −80 °C.

The ExbB–ExbD complexes (ExbB–ExbD, ExbB$_{C25S}$–ExbD$_{D10C}$, ExbB$_{C25S}$–ExbD$_{L40C}$, ExbB$_{C25S}$–ExbD$_{P50C}$) were expressed, solubilized with

*n*-Dodecyl β-D-maltoside (DDM) and purified by affinity chromatography as previously described[5]. For the crosslinking analysis, SEC experiments were performed on a Superose6 increase 10/30 column (GE Healthcare) in PBS imidazole 50 mM DDM 0.1% w/v.

**Nanodisc reconstitution.** *E. coli* polar lipid extract (Avanti Polar) dissolved in chloroform was dried in a glass tube using a nitrogen stream. The dried lipids were resuspended in Tris 20 mM pH7.5 NaCl 150 mM cholate 200 mM for a final concentration of 100 mM lipid.

Affinity purified ExbB–ExbD was reconstituted with MSP1E3D1 and *E. coli* polar lipid extract in a final ratio of 1:2:160, in PBS imidazole 20 mM cholate 10 μM DDM 0.1% and allowed to incubate on ice for several hours. Detergent was then removed with the addition of 0.1 gr/ml wet bio-beads SM2 (Bio-Rad) and gentle agitation at 4 °C overnight. The bio-beads were removed by filtration and aggregated material was pelleted by ultracentrifugation. The empty nanodiscs were removed with an additional affinity chromatography step using a HisTrapHP 5 ml column (GE Healthcare) in Tris 25 mM pH7.4 NaCl 200 mM. The ExbB–ExbD–MSP1E3D1 complexes were eluted with the same buffer containing 250 mM imidazole. The complex was then injected on a Superose6 increase 10/30 (GE Healthcare) in Tris 25 mM pH7.4 NaCl 200 mM. The fraction centered on the main peak was concentrated by ultrafiltration and used for cryoEM.

**EM sample preparation and data acquisition.** Three microliters of the sample at 2 mg/ml was applied to a Quantifoil R1.2/1.3 Cu 200 mesh grid (Electron Microscopy Sciences, Protochips, Inc.) that had been glow discharged for 30 s at 30 mA (Pelco easiGlow, Ted Pella, Inc.), blotted, and then immediately plunged into liquid ethane using a Vitrobot system (Thermo Fisher Scientific, Inc.). The freezing conditions were as follows: 100% humidity, temperature 22 °C, 15 s wait time, 4–6 s blot time and 0 to +2 blot force.

A first data collection was performed with a FEI Titan Krios operating at 300 keV coupled with a Gatan K2 direct electron detector via the Latitude software (Gatan, Inc) at the Center for Molecular Microscopy (NCI). Micrographs were collected as dose-fractionated movies at a ×14,000 nominal magnification resulting in a pixel size of 1.72 Å/pixel in the counting mode, with 400 ms exposure per frame and 38 frames per movie. A total of 2917 movies were collected. The total dose in the EM data collection was 40 e−/Å$^2$ per movie. The nominal defocus range used was −1.2 to −2.5 μm.

The second data collection was performed with a Titan Krios G3 cryo-electron microscope (Thermo-Fisher) operated at 300 kV at the MICEF (NIH). Micrographs were recorded as dose-fractionated movies with a Gatan K2 Summit direct electron detector operated in counting mode at a nominal magnification of ×130,000 (calibrated pixel size of 1.06 Å on the sample level). A Gatan Imaging Filter (GIF) Quantum LS was installed with the K2 Summit camera and the slit width was set to 20 eV. The dose rate on the camera was set to ~8 e⁻/pixel/s. The total exposure time of each movie was 10 s fractionated into 50 frames with 0.2 s exposure time for each frame. A total of 5244 movies were acquired with automation using Leginon[17].

**Image processing**. The movie frames from the first data collection were gain corrected and aligned using MotionCor2[18], and defocus determination was performed on averaged images with CTFFIND4[19]. A total of 1,867,538 particles were picked using the convolution neural network option for particle picking in EMAN2[20,21]. Particles were extracted as 128 × 128 pixels boxes and processed using RELION3[22]. 543,898 particles were selected after three rounds of 2D classification. A low resolution de novo 3D initial model was prepared using the stochastic gradient descent (SGD) algorithm and used for 3D classification into five classes with C1 symmetry. The best resulting class represented 37.8% of the selected particles (205,777 out of 534,898) and showed the best resolution (7.34 Å). The particles for this class were further 3D autorefined with C1 symmetry, generating a 5.4 Å resolution structure after postprocessing with a soft mask, using the Fourier Shell Correlation (FSC) gold-standard criterion at 0.143.

Data processing for the second dataset was performed as follows. All frames of each movie were aligned for correction of beam-induced drift using MotionCor2[18]. Two average images were generated by MotionCor2 from each movie: one with dose weighting and the other one without. The average images without dose weighting were only used for defocus determination, which was carried out by CTFFIND4[19]. Quality of the micrographs was evaluated using the results from CTFFIND4. A set of 4449 micrographs, whose defocus values were between −0.7 to −2.5 μm and max resolution was better than 4.5 Å, were selected for further processing. The single particle data analysis was performed following the standard procedures in RELION3[22] with few modifications as summarized in Supplementary Fig. 1. A total of 3,206,108 particles in the dose weighted average images were picked with Gautomatch (https://www.mrc-lmb.cam.ac.uk/kzhang/Gautomatch/) using the result from the previous cryo-EM 3D reconstruction as the template. The particles extracted in 160 × 160 pixels and then 2x binned in 80 × 80 pixels (pixel size 2.12 Å) for more efficient initial processing. The particles were classified by 2D classification and particles in those classes with interpretable structural features in their 2D class averages were selected. This was repeated twice with careful selection. A set of 1,891,941 particles were selected and sent to 3D classification for five classes with C5 symmetry using the result from the previous cryoEM 3D reconstruction as the starting model. The particles were separated into five subsets by 3D classes and each subset was further classified using 2D classification. The good particles from each 2D classification process were combined to generate a set of 1,310,001 particles. These particles were subjected to another 3D classification for five classes with C1 symmetry. A set of 1,123,304 particles were selected from two 3D classes that possessed well resolved sausage-like features of α-helices. This set of particles was then used for 3D autorefinement with C5 symmetry. The result was used to re-center the particles as well as to remove overlapping particles. Therefore, 1,118,493 particles were re-extracted with 192 × 192 pixels without binning (pixel size 1.06 Å). 2D classification was used to remove bad particles and generated a set of 1,018,126 particles. These particles were sorted into four classes by 3D classification with C1 symmetry for 50 iterations. The beginning 30 iterations and following 20 iterations were with 7.5° and 3.7° angular sampling intervals, respectively. These four classes all showed densities of two helices in the pore of ExbB pentamer (Supplementary Fig. 2), suggesting that nearly all particles in the sample have a similar structure with 2 ExbD helices in the pore. The differences between these classes are primarily from the region of lipid nanodisc. The particles were then subjected to another similar 3D classification with some modifications (three classes with C5 symmetry). 544,066 particles in one class with best resolved features of ExbB as well as densities from tightly bound membrane scaffold proteins were selected for further processing. These particles were used for 3D autorefinement with C5 symmetry that generated reconstruction at 3.7 Å resolution. This result was then used to facilitate polishing of the particles. The polished particles were screened by 2D classification and 512,927 particles were selected. 3D autorefinement using the selected polished particles with C5 symmetry showed an improvement of resolution to 3.4 Å. These particles were also sorted into three classes by 3D classification with C1 symmetry for 70 iterations. The beginning 30 iterations and following 20 iterations were with a spherical mask and angular sampling interval decreasing from 7.5° to 3.7°. Interactions 51–70 were performed with a carefully constructed soft mask that only included ExbB–ExbD and not the lipid nanodisc. The best class, with 153,974 particles, clearly resolved the pitch of α-helices as well as some bulky side chains, while the other two classes showed poor resolution, likely because those particles failed to be correctly aligned when the densities from the lipid nanodisc were not considered. The particles in the best class were used in 3D autorefinement with C1 symmetry and the soft mask to obtain a reconstruction at 3.7 Å resolution. The defocus values of these particles were refined (Ctf refinement), which improved the

resolution of a similar 3D autorefinement to 3.6 Å. The particles were subjected to polishing one more time and selected by 2D classification to generate a set of 147,738 particles. This additional particle polishing improved the resolution of 3D autorefinement to 3.5 Å. The result was used for a final 3D classification for three classes with C1 symmetry, the soft mask, and 30 iterations of local search (0.9° angular sampling interval, 3° angular search range, 0.4 pixel search step, and 3 pixel offset search range). The best class selected 85,936 particles, which were then used for final 3D refinement using both RELION3 and cryoSPARC2[23]. The 3D autorefinement with C1 symmetry and the soft mask using RELION3 generated a reconstruction at 3.5 Å. The non-uniform refinement with C1 symmetry and a dynamic mask using cryoSPARC2 obtained a similar structure at 3.3 Å resolution. The resolution calculation was based on the 'gold-standard' FSC at 0.143 criterion with automatic soft mask. The cryoEM maps were sharpened with B-factor and low-pass filtered using the *relion_postprocess* program in RELION3 or the *Local Filtering* function in cryoSPARC2. The local resolution was calculated by ResMap[24] or by cryoSPARC2 with an algorithm similar to the *blocres* program[25] using two cryoEM maps independently refined from halves of data. CryoEM data collection, refinement, and validation statistics are given in Table S1.

**Model building and refinement**. Model building into the ExbB–ExbD 3.5 Å map was started by fitting the pentameric ExbB crystal structure (PDB:5SV0[5]) into the map with UCSF Chimera[26] and real-space refining it in Phenix[27]. The resulting structure was rebuilt in Coot[28] using the more detailed, Phenix auto-sharpened 3.3 Å map. Two ExbD molecules were built into the map in Coot, based on the TM helix of the ExbD crystal structure (PDB:5SV1[5]). Lipids were individually built in Coot then the whole structure was real-space refined in Phenix.

**Reporting summary**. Further information on research design is available in the Nature Research Reporting Summary linked to this article.

## Data availability
Atomic coordinates and structure factors for ExbBD have been deposited in the EMDB and wwPDB under accession codes EMD-20583 and PDB 6TYI. Source data for all figures and files are available from the authors upon reasonable request.

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

## Acknowledgements
The authors thank Huaibin Wang and Haifeng He for technical support on the NIH MICEF Titan Krios Electron Microscope, and Bastien Serrano at the IMM for help with cloning. H.C., I.B, and S.K.B. are supported by the Intramural Research Program of the NIH, NIDDK. X.N. and J.J. are supported by the Intramural Research Program of the NIH, NHLBI. R.L. is supported by the Centre National de la Recherche Scientifique, the Aix-Marseille Université and grants from the Agence Nationale de la Recherche (ANR-14-CE09-0023 and ANR-18-CE11-0027). The EM part of the work performed at the Center for Molecular Microscopy (CMM) was funded by FNLCR Contract HHSN261200800001E. This work utilized the NIH Multi-Institute Cryo-EM Facility (MICEF) and the computational resources of the NIH HPC Biowulf cluster (http://hpc.nih.gov).

## Author contributions
H.C. expressed and purified the proteins, prepared nanodisc samples, prepared EM grids, collected and analyzed data, and wrote the manuscript. I.B. collected and analyzed data, constructed the model, and wrote the manuscript. T.F. and N.D.V. prepared EM grids and collected data. X.N. and J.J. collected and processed the EM data, and wrote the manuscript. R.L. designed the cysteine substitutions and commented on the manuscript. S.K.B. directed the work and wrote the manuscript.

## Competing interests
The authors declare no competing interests.
