## [Peer Review File · Communications Biology]

Reviewers' comments:

Reviewer #1 (Remarks to the Author):

The bacterial Ton complex harvests energy from the proton gradient across the inner membrane to transmit it to specialized transporters in the outer membrane. In their paper "Cryo-EM structure of the Ton motor subcomplex in lipid nanodiscs", Celia et al. provide a high-resolution cryo-EM structure of the ExbB-ExbD subunits of the TonB-ExbB-ExbD molecular motor. They observe a pentameric pore structure of ExbB with a central pair of ExbD helices. The overall results contradict an earlier model of the same group based on a crystal structure where only a single ExbD helix was observed inside a pentameric ExbB pore. Still, the current results are in good agreement with previous biochemical results, and are supported by additional site-specific crosslinking experiments. The authors suggest that the previous crystal structure may have been affected by the absence of a dimerization domain of ExbD, which was included in the current cryoEM study (but remains disordered), but might also include a discussion of different lipid(-mimicking) environments. The EM analysis and crosslinking studies have been carried out carefully and fully support the conclusions drawn here. The updated model is critically important for a mechanistic understanding of Ton function, but leaves most relevant questions unanswered, e.g. the detailed dynamics of the motor or the coupling mechanism to proton translocation.

Overall the findings of this paper are relevant to the field and the format chosen is fully appropriate for reporting the current data. The manuscript should be published after minor revisions addressing the points below.

Text:

- No reference to figure 2F
- Include discussion on relevance of lipid environment or detergent solubilization

Figure F2:

- Labels of side chains are very small
- Unit for y axis in sec elution profile
- Indicating observed or suggested helix motions also in a figure might be helpful for the reader

Structure validation:

- The number of atoms differs between supplement and validation report
- The number of residues differs between supplement and validation report
- Clash score is quite high and appears to include in particular ligand clashes. Improving ligand modelling might improve clash score

Reporting summary

- Cryosparc's latest version is 2.8, so using cryosparc3 seems unlikely

Reviewer #2 (Remarks to the Author):

The Ton complex is a molecular motor located on the inner membrane of Gram-negative bacteria. The authors use single particle cryo-electron microscopy to determine the structure of the full length ExbB-ExbD subcomplex reconstituted into nanodiscs. The reported structure shows a pentamer of ExbB that encloses a dimer of ExbD. In their previous work the authors had confirmed this stoichiometry,

however, at the time they proposed that only one copy of ExbD is found within the pore, with the other copy being located on the outside of the ExbB pentamer.

Reconstitution into nanodiscs is a powerful approach for studying small membrane proteins. This work has potential; however, I would like to encourage the authors to critically reexamine their cryo-EM data processing.

Comments for the authors:

1. For the Cryo-EM data processing, the authors mention that the automated particle picking was performed in Gautomatch, where the input was projections of a pre-existing cryo-EM 3D reconstruction as template. However, it is not clear where that 3D reconstruction came from and if it is from the current data collection or completely independent data and if anything had been done to the EM map. Ideally, the initial particle picking should be without a template and subsequently, 2D classes or projections from the first 3D classification/refinement could be used to re-pick particles.
2. Next, the authors are again using as a starting initial model an EM map the origin of which is not clear (Online methods, line 66). To make sure that the data processing is not biased, the authors should use the de novo initial model generation in Relion3 (or another software) to create a low-resolution 3D map that can be then used for 3D classification.
3. In Supplementary Figure 1, class 2 and 3 from the initial 3D classification appear to have different handedness. Could the authors double-check whether that is the case? Or provide evidence that the handedness is the same? Perhaps the authors can use their crystal structure to help determine that. If the handedness is indeed different, those particles should be discarded at this stage or alternatively, the hand of the particles can be flipped.
4. Additionally, after the first 3D classification, it seems that the first four classes were chosen for further processing. Perhaps the authors should consider, at this stage, to focus on the best resolved class first.
5. Given the results and interpretation from their previous work and the new conclusions presented in this manuscript, it may be worthwhile for the authors to go back and explore whether the cryo-EM data shows evidence of variation of the pore opening and/or the number ExbD domains within the pore. For example, the 3D classification results indicate that there might be some structural variations (height, width of classes 2 and 3).
6. Additionally, being more selective early on (at least initially) might help simplify the data processing workflow. It is not clear why the authors go through multiple rounds of 2D and 3D classification, especially after polishing.
7. The authors alternate between using C1 and C5 symmetry during 3D classification and it is not clear why.
8. Have the authors considered doing a complete refinement in cryoSPARC for all picked particles and not just at the very last step? This could serve as validation of the EM map.
9. Could the authors include some additional information about the complex such as molecular weight and dimensions and more background about the individual components? Supplementary Figure 4 could benefit from adding dimensions information.
10. In Figure 2A the text is very difficult to read (size and color).

11. Can the authors show an overlay of their crystal structure onto the EM map?

Comments to Reviews:

We thank the reviewers for their insightful comments, which we have attempted to address in full.

Reviewer #1 (Remarks to the Author):

The bacterial Ton complex harvests energy from the proton gradient across the inner membrane to transmit it to specialized transporters in the outer membrane. In their paper “Cryo-EM structure of the Ton motor subcomplex in lipid nanodiscs”, Celia et al. provide a high-resolution cryo-EM structure of the ExbB-ExbD subunits of the TonB-ExbB-ExbD molecular motor. They observe a pentameric pore structure of ExbB with a central pair of ExbD helices. The overall results contradict an earlier model of the same group based on a crystal structure where only a single ExbD helix was observed inside a pentameric ExbB pore. Still, the current results are in good agreement with previous biochemical results and are supported by additional site-specific crosslinking experiments. The authors suggest that the previous crystal structure may have been affected by the absence of a dimerization domain of ExbD, which was included in the current cryoEM study (but remains disordered) but might also include a discussion of different lipid(-mimicking) environments. The EM analysis and crosslinking studies have been carried out carefully and fully support the conclusions drawn here. The updated model is critically important for a mechanistic understanding of Ton function, but leaves most relevant questions unanswered, e.g. the detailed dynamics of the motor or the coupling mechanism to proton translocation. Overall the findings of this paper are relevant to the field and the format chosen is fully appropriate for reporting the current data. The manuscript should be published after minor revisions addressing the points below.

Text:

- No reference to figure 2F
- Include discussion on relevance of lipid environment or detergent solubilization

Figure 2F is now referenced (line 66), and a paragraph has been added that discusses the relevance of lipid environment (lines 81-85):

“Because of the reconstitution in lipid nanodiscs, the ExbB-ExbD subcomplex is observed in a natural lipid environment. The cryo-EM map reveals densities that correspond to phospholipids closely associated with the ExbB pentamer. Four phospholipids were built, three PE and one PG, and are part of the cytoplasmic leaflet of the membrane. The crystal structure did not reveal any associated lipids, likely because they were displaced by the detergent.”

Figure F2:

- Labels of side chains are very small
- Unit for y axis in sec elution profile
- Indicating observed or suggested helix motions also in a figure might be helpful for the reader

The labels of Figure 2 have been resized, and the y axis for the sec elution profile has been added (milli-absorbance units at 280nm).

We do not observe helix motions, but rather structural differences between the cryo-EM and crystallographic structures. The superposition of the two structures in the Supplementary Fig 4 a-b shows these differences, as well as the two supplementary movies. If the reviewer thinks we should include another figure or annotate the existing Sup Fig 4 a-b, we will be happy to do so.

Structure validation:

- The number of atoms differs between supplement and validation report
- The number of residues differs between supplement and validation report
- Clash score is quite high and appears to include in particular ligand clashes. Improving ligand modelling might improve clash score

The number of atoms and residues have been updated in the Sup Table 1.

About the clash score, the ligands were built and geometry optimized with Elbow (Phenix). Ligands were modeled only into density with clear connectivity. There is additional tubular density for lipid chains but they do not make up full lipid molecules. Since this density is ambiguous it was not modeled. We think that the additional density slightly distorts the ligands in the refinement, yielding clashes. We have tried to further refine the ligands but the statistics did not significantly improve.

Reporting summary

- Cryosparc's latest version is 2.8, so using cryosparc3 seems unlikely

Cryosparc3 has been replaced with cryosparc2 in the reporting summary.

Reviewer #2 (Remarks to the Author):

The Ton complex is a molecular motor located on the inner membrane of Gram-negative bacteria. The authors use single particle cryo-electron microscopy to determine the structure of the full length ExbB-ExbD subcomplex reconstituted into nanodiscs. The reported structure shows a pentamer of ExbB that encloses a dimer of ExbD. In their previous work the authors had confirmed this stoichiometry, however, at the time they proposed that only one copy of ExbD is found within the pore, with the other copy being located on the outside of the ExbB pentamer.

Reconstitution into nanodiscs is a powerful approach for studying small membrane proteins. This work has potential; however, I would like to encourage the authors to critically reexamine their cryo-EM data processing.

Comments for the authors:

1. For the Cryo-EM data processing, the authors mention that the automated particle picking was performed in Gautomatch, where the input was projections of a pre-existing cryo-EM 3D reconstruction as template. However, it is not clear where that 3D reconstruction came from and if it is from the current data collection or completely independent data and if anything had been done to the EM map. Ideally, the initial particle picking should be without a template and subsequently, 2D classes or projections from the first 3D classification/refinement could be used to re-pick particles.

We now have added in the methods sections the data collection and image processing that yielded the 3D reference used for particle picking and 3D initial model.

lines 41-49 for data collection:

"A first data collection was performed with a FEI Titan Krios operating at 300 keV coupled with a Gatan K2 direct electron detector via the Latitude software (Gatan, Inc) at the Center for Molecular Microscopy

(NCI). Micrographs were collected as dose-fractionated movies at a 14,000× nominal magnification resulting in a pixel size of 1.72 Å/pixel in the counting mode, with 400ms exposure per frame and 38 frames per movie. A total of 2917 movies were collected. The total dose in the EM data collection was 40 e⁻/Å² per movie. The nominal defocus range used was -1.2 to -2.5 μm. The second data collection was performed with a Titan Krios G3 cryo electron microscope (Thermo-Fisher) operated at 300 kV at the MICEF (NIH).”

lines 59-70 for image processing:

“The movie frames from the first data collection were gain corrected and aligned using MotionCor2⁵, and defocus determination was performed on averaged images with CTFFIND4⁶. A total of 1,867,538 particles were picked using the convolution neural network option for particle picking in EMAN2⁷. Particles were extracted as 128x128 pixels boxes and processed using RELION3⁸. 543,898 particles were selected after three rounds of 2D classification. A low resolution de novo 3D initial model was prepared using the stochastic gradient descent (SGD) algorithm and used for 3D classification into 5 classes with C1 symmetry. The best resulting class represented 37.8% of the selected particles (205,777 out of 534,898) and showed the best resolution (7.34Å). The particles for this class were further 3D autorefined with C1 symmetry, generating a 5.4Å resolution structure after postprocessing with a soft mask, using the Fourier Shell Correlation gold standard criterion at 0.143. Data processing for the second dataset was performed as follow.”

2. Next, the authors are again using as a starting initial model an EM map the origin of which is not clear (Online methods, line 66). To make sure that the data processing is not biased, the authors should use the de novo initial model generation in Relion3 (or another software) to create a low-resolution 3D map that can be then used for 3D classification.

As mentioned in the added text for image processing, an unbiased de novo 3D initial model was prepared with the first dataset.

3. In Supplementary Figure 1, class 2 and 3 from the initial 3D classification appear to have different handedness. Could the authors double-check whether that is the case? Or provide evidence that the handedness is the same? Perhaps the authors can use their crystal structure to help determine that. If the handedness is indeed different, those particles should be discarded at this stage or alternatively, the hand of the particles can be flipped.

The reviewer is right that the initial 3D classification resulted in different handedness. EM images are 2D projections of 3D objects and do not provide the absolute handedness. Therefore there is a 50% chance to get a flipped handedness unless additional information is available, such as a tilt series of EM images. Alternatively, the handedness can be determined later when the 3D reconstruction reaches a high resolution to reveal some features of handedness, such as the right-handed α-helices. In our case, we have determined there is only one handedness and the different ones in the initial 3D classification resulted from random flipping of handedness at the beginning of 3D classification when the resolution was low. Therefore, it is not necessary to discard the particles in the 3D classes with flipped handedness or flip the hand of those particles.

4. Additionally, after the first 3D classification, it seems that the first four classes were chosen for further processing. Perhaps the authors should consider, at this stage, to focus on the best resolved class first.

The reviewer is right that it is a common practice that only the best resolved 3D classes are chosen for further processing. Because we had to deal with symmetry mismatch (2 helices in a pore with a pseudo 5-fold symmetry) and subtle conformational changes, we chose not to discard particles too aggressively at the beginning to avoid losing valuable information in those classes with smaller populations of particles.

5. Given the results and interpretation from their previous work and the new conclusions presented in this manuscript, it may be worthwhile for the authors to go back and explore whether the cryo-EM data shows evidence of variation of the pore opening and/or the number ExbD domains within the pore. For example, the 3D classification results indicate that there might be some structural variations (height, width of classes 2 and 3).

The reviewer made an excellent suggestion of analyzing the variations of the pore. Actually, we had done this carefully. By including almost all the particles (1,018,126 particles) in the well resolved 3D classes, elaborated 3D classification showed that there are no significant variations in the structure of the pore or the number of ExbD domains and the majority of variations among the different classes are from the micelle of the lipid nanodisc (Supplementary Figure 2).

6. Additionally, being more selective early on (at least initially) might help simplify the data processing workflow. It is not clear why the authors go through multiple rounds of 2D and 3D classification, especially after polishing.

We had tried a simple data processing workflow but it did not give us an optimal result. The reason is because the symmetry mismatch (2 helices in a pore with a pseudo 5-fold symmetry) and the subtle difference between one and two ExbD domains (1 vs. 2 helices in the pore) are difficult problems. Therefore, we were less selective initially to avoid losing valuable information and used multiple rounds of 2D and 3D classification to gradually separate good/bad particles and different conformations. Indeed, this approach worked well for these challenging problems.

The 2D classifications after particle polishing were to remove the particles on the areas of the electron detector with defective pixels. Because RELION does not remove defective pixels during particle polishing, they become ripples in the polished images (after Fourier transform, modulation, and reverse Fourier transform).

The 3D classifications after particle polishing were to select the most homogeneous particles for 3D reconstruction to a higher resolution. Polished particles have enhanced high-resolution signals which can be useful to separate small differences.

7. The authors alternate between using C1 and C5 symmetry during 3D classification and it is not clear why.

The C5 symmetry was used throughout the 3D classifications and 3D refinement before particle polishing, with the only exception for the second round of 3D classification. Because the pore has a pseudo 5-fold symmetry, the C5 symmetry was used to effectively separate “good” particles from “bad” particles using 3D classification. After particle polishing, the C1 symmetry was used for both 3D classification and 3D refinement to reveal the asymmetric structure of the complex.

8. Have the authors considered doing a complete refinement in cryoSPARC for all picked particles and not just at the very last step? This could serve as validation of the EM map.

Because cryoSPARC does not allow fine-tuned controls on image classification, we were not able to find a feasible solution to deal with the symmetry mismatch problem using cryoSPARC. In the meantime, we had been carefully validating the results at the key steps during processing using RELION (e.g. Supplementary Figure 2) and the quality of the EM map is satisfactory at the claimed resolution (Supplementary Figure 5).

9. Could the authors include some additional information about the complex such as molecular weight and dimensions and more background about the individual components? Supplementary Figure 4 could benefit from adding dimensions information.

Information about the Ton components have been added (lines 23-27), and the dimensions of the complex are shown on Sup Fig 4 e.

Lines 23-27:

“TonB (239 residues) and ExbD (141 residues) have a similar topology: a short N-terminal domain in the cytoplasm, followed by a single transmembrane (TM) helix, and a periplasmic domain composed of a flexible linker and a folded C-terminal domain²⁻⁴. ExbB (244 residues) has three TM helices, with a short periplasmic N-terminal domain, a small periplasmic loop between TM 2 and 3, and a large cytoplasmic domain⁵.”

10. In Figure 2A the text is very difficult to read (size and color).

The text in Fig2 has been resized, and the color used for the two “Asp25” has been modified.

11. Can the authors show an overlay of their crystal structure onto the EM map?

An overlay of the crystal structure onto the EM map has been added in Sup Fig 4 e-f.

REVIEWERS' COMMENTS:

Reviewer #1 (Remarks to the Author):

The authors have fully addressed all comments raised by the reviewers. The technical description of EM data processing is now fully adequate, formatting issues are resolved, and an additional figure panel in the Supplement has been provided as requested.

Reviewer #3 (Remarks to the Author):

Thank you to the authors for thoroughly addressing all the points. Only a few minor notes to consider:

Figure 1A: the label "membrane" appears cut, additionally the gray color is too faint in this figure but also in the other figures as well.

Figure 2A: Labels are better but still faint; 2 E,F could benefit from larger/thicker stroke labels.

Supplementary Figure 4 E,F: Thank you for including the crystal structure. With the current color scheme, it is difficult to see the structure, maybe a different color of the crystal structure and/or mesh representation of the EM map might help. Labels could be larger (as mentioned above)

Online methods

Line 44: Please double-check the microscope magnification and pixel size

Image processing: Please consider referencing Bell JM et al, JSB 2018 for the neural network particle picking in EMAN2 (line 62)

Line 70: "as follow" to "as follows"

Line 80: instead of "a previous cryo-EM 3D reconstruction", please correct to "the previous cryo-EM 3D reconstruction" to make it clear that you are referring to the structure obtained from the initial data set.

Comments to Reviews:

We thank the reviewers for their insightful comments, which we have attempted to address in full.

REVIEWERS' COMMENTS:

Reviewer #1 (Remarks to the Author):

The authors have fully addressed all comments raised by the reviewers. The technical description of EM data processing is now fully adequate, formatting issues are resolved, and an additional figure panel in the Supplement has been provided as requested.

Wonderful, thank you.

Reviewer #3 (Remarks to the Author):

Thank you to the authors for thoroughly addressing all the points. Only a few minor notes to consider:

Figure 1A: the label “membrane” appears cut, additionally the gray color is too faint in this figure but also in the other figures as well.

We corrected figures 1 and 2 as directed.

Figure 2A: Labels are better but still faint; 2 E,F could benefit from larger/thicker stroke labels. Also corrected in new Figure 2.

Supplementary Figure 4 E,F: Thank you for including the crystal structure. With the current color scheme, it is difficult to see the structure, maybe a different color of the crystal structure and/or mesh representation of the EM map might help. Labels could be larger (as mentioned above)

We used a mesh representation of the EM map to better show the crystal structure.

Online methods

Line 44: Please double-check the microscope magnification and pixel size checked and verified.

Image processing: Please consider referencing Bell JM et al, JSB 2018 for the neural network particle picking in EMAN2 (line 62)

Thank you – these are now referenced.

Line 70: “as follow” to “as follows” corrected.

Line 80: instead of “a previous cryo-EM 3D reconstruction”, please correct to “the previous cryo-EM 3D reconstruction” to make it clear that you are referring to the structure obtained from the initial data set.

corrected.